# Identifying Coastal Wetlands Changes Using a High-Resolution Optical Images Feature Hierarchical Selection Method

**Ruijuan Wu [1,2,3,*] and Jing Wang [4]**

1    Key Laboratory for Digital Land and Resources of Jiangxi Province, East China University of Technology, Nanchang 330013, China
2    School of Geography and Resource Science, Neijiang Normal University, Neijiang 641102, China
3    State Key Laboratory of Resources and Environmental Information System, Institute of Geographic Sciences and Natural Resources Research, CAS, Beijing 100101, China
4    College of Water Sciences, Beijing Normal University, Beijing 100875, China
*    Correspondence: rjwu@njtc.edu.cn

**Abstract:** Coastal wetlands are dynamic and fragile ecosystems where complex changes have taken place. As they are affected by environmental changes and human activities, it is of great practical significance to monitor coastal wetlands changes regularly. High-resolution optical data can observe changes in coastal wetlands, however, the impact of different optical features on the identification of changes in coastal wetlands is not clear. Simultaneously, the combination of many features could cause the "dimension disaster" problem. In addition, only small amounts of training samples are accessible at pre- or post-changed time. In order to solve the above problems, the feature hierarchical selection method is proposed, taking into account the jumping degree of different image features. The influence of different optical features on wetland classification was analyzed. In addition, a training samples transfer learning strategy was designed for wetland classification, and the classification result at pre- and post-changed times were compared to identify the "from-to" coastal wetlands changes. The southeastern coastal wetlands located in Jiangsu Province were used as a study area, and ZY-3 images in 2013 and 2018 were used to verify the proposed methods. The results show that the feature hierarchical selection method can provide a quantitative reference for optimal subset feature selection. A training samples transfer learning strategy was used to classify post-changed optical data, the overall accuracy of transferred training samples was 91.16%, and it ensures the accuracy requirements for change identification. In the study area, the salt marsh increased mainly from the sea area, because salt marshes expand rapidly throughout coastal areas, and aquaculture ponds increased from the sea area and salt marshes, because of the considerable economic benefits of the aquacultural industry.

**Keywords:** change detection; feature selection; jumping degree; transfer learning; coastal wetlands





## 1. Introduction

Coastal wetlands, which are located in the interactive zone between terrestrial and aquatic ecosystems, are dynamic and fragile ecosystems [1]. They play important roles in water conservation, regional climate regulation, flood control, biodiversity protection, and so on [2,3]. Over the past century, coastal wetlands have experienced obvious degradation or even disappearance, as a result of environmental changes and human activities, such as climate change, urban expansion, livestock grazing, and agricultural development [4]. Change detection (CD) is the appropriate way to observe wetland changes, therefore accurate and timely CD is fundamental to reliable wetland management and successful wetland protection.

Optical images [5–7] and synthetic aperture radar (SAR) images [8–10] have been effectively utilized for dynamic monitoring in wetlands. Spectral-enhanced features [11], textural features [12], spectral-spatial features [13], spectral-textural features [14], and

spectral-spatial-textural features [15,16] have been utilized for change detection, however, there is inevitably redundancy among these features. If all of these features are input into the image classifier, on the one hand, it is easy to cause the "dimension disaster" problem, and computer operation becomes slow. On the other hand, the addition of some features is not conducive to the improvement of image classification and change detection. Therefore, it is necessary to select some features to obtain optimal feature subsets.

Feature selection methods select optimal feature subsets from the available image features, which are divided into filter methods and wrapper methods [17]. Recently, principal component analysis [18], random forest-based [19], genetic particle swarm optimization [17], sequential forward selection [20], ensemble learning [21], and adaptive feature selection network [22] have been proposed for feature selection. Random forest is one of the wrapper methods and it removes irrelevant features one by one in order to obtain optimal feature subsets. When there are hundreds or thousands of features, it will take more time to remove irrelevant features one by one, therefore, it is necessary to improve the efficiency of feature selection using random forest.

In addition, due to the large-scale and complex geographical conditions of coastal wetlands [23], it is time-consuming and laborious to collect training samples. According to the availability of training samples, CD methods can be divided into two categories: supervised CD and unsupervised CD. In addition, according to the final detailed requirements for the changed regions, CD can be divided into two categories: binary CD and multiple CD. Binary CD only separates changed areas from unchanged areas, while multiple CD can not only extract the changed areas but also obtain the "from-to" change information. A number of binary CD methods have been proposed, such as change vector analysis [24], clustering method [25], threshold method [26], and deep learning [27]. The development of binary CD methods is summarized in references [28–31].

Compared with binary CD, multiple CD is more challenging. Some approaches have been developed, such as post classification comparison [32], CD in change vector analysis polar domain [24], and slow feature analysis [33]. However, these methods have been developed based on the assumption that there are either some or no ground reference samples at both pre- and post-changed times. Only small amounts of ground reference data are available at only one of the pre- and post-changed times [34], therefore, how to utilize limited ground reference samples for multiple CD is another important consideration.

The contribution of this article is to propose a random forest-based feature hierarchical selection method in order to obtain optimal feature subsets, and these optimal feature subsets could then be utilized for coastal wetlands classification. In addition, a simple training samples transfer learning strategy was designed for identifying coastal wetland changes, where small amounts of ground training samples are available only at the pre-changed time. The feature optimization can improve the accuracy of coastal wetlands classification, and samples transfer learning can be more suitable for the actual monitoring of changes in wetlands.

The rest of this article is organized as follows. The study area and data descriptions are presented in Section 2, the proposed feature hierarchical selection and designed changes identification using sample transfer learning are described in Section 3, our experimental results are reported in Section 4, and conclusions are summarized in Section 5.

## 2. Materials

### 2.1. Study Area

The study area was a typical coastal wetland and is located in Dongtai city in the southeastern part of the Jiangsu province in China (Figure 1). The wetland types in the region are sea, open water, farmland, aquaculture pond, salt marsh, and building.

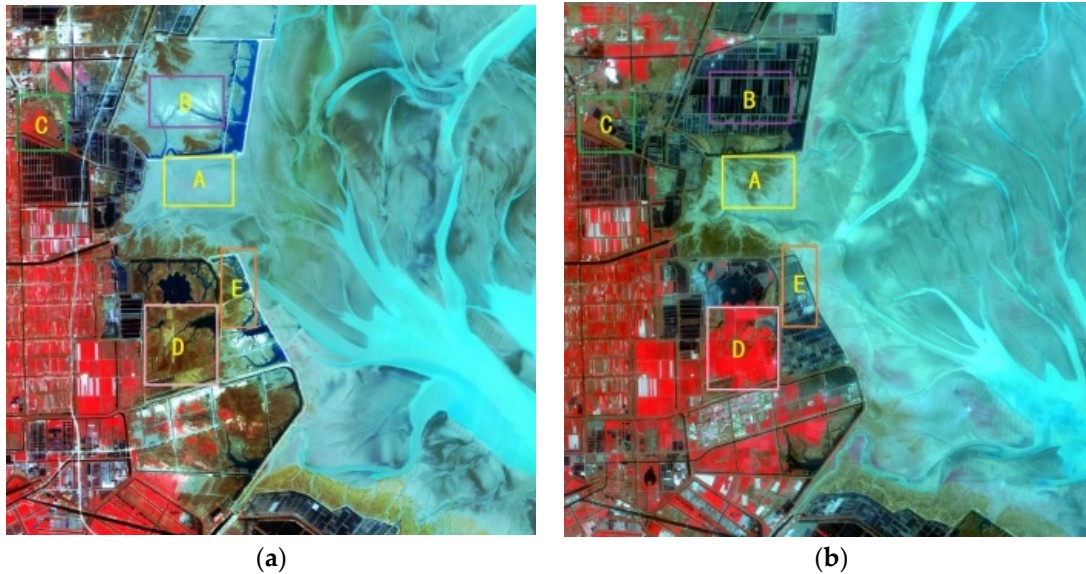

(**a**)                    (**b**)

**Figure 1.** Experimental data: (**a**) ZY-3 image in 2013; (**b**) ZY-3 image in 2018.

In the study area, the change patterns consisted of Se–SM (the change from sea to salt marsh marked in rectangle A), Se–AP (the change from sea to aquaculture pond marked in rectangle B), FL–AP (the change from farmland to aquaculture pond marked in rectangle C), SM–FL (the change from salt marsh to farmland marked in rectangle D) and SM–AP (the change from salt marsh to aquaculture pond marked in rectangle E).

*2.2. Dataset*

High spatial-resolution Ziyuan-3 (ZY-3) images in 2013 and 2018 were used as experimental data, the acquisition dates and spatial resolutions are listed in Table 1. Image preprocessing included radiometric correction, image registration, and image cropping using ENVI 5.6 software.

**Table 1.** Multi-temporal optical data.

| Sensor | Acquisition Date | Spatial Resolution |
|---|---|---|
| ZY-3 | 4 March 2013 | 5.8 |
| ZY-3 | 22 March 2018 | 5.8 |

## 3. Methods

*3.1. Feature Extraction*

Different feature types were extracted from ZY-3 images, including spectral-based, texture-based, morphological-based, transform-based, edge-based, and vegetation indexes.

(a)   Spectral-based features contained five spectral features (blue, green, red, near-infrared bands, and brightness).

(b)   Texture-based features were GLCM (gray-level co-occurrence matrix)-based features, including mean, variance, homogeneity, dissimilarity, contrast, entropy, angular second moment, and correlation. GLCM-based textures were influenced by the window size of images, so six different window sizes, $3 \times 3$, $7 \times 7$, $11 \times 11$, $15 \times 15$, $19 \times 19$, and $23 \times 23$, were chosen to calculate GLCM textures. For four spectral bands (blue, green, red, and near-infrared band), 192 GLCM-based textures were extracted.

(c)   Morphological profiles (MPs) and differential morphological profiles (DMPs) can express the morphological characteristics of land cover, so the opening and closing of both MPs and DMPs were used to extract the morphological-based features. Five different scales [1–5] were chosen to describe the morphological characteristics on a

fine to coarse scale. Eighty morphological-based features were extracted from four spectral bands.

(d) Transform-based features were extracted using non-subsampling shearlet transform (NSST). NSST decomposition was used to obtain high-frequency sub-bands and low-frequency sub-bands, and NSST reconstruction was used to reconstruct image features. In order to describe the above features on a coarse to fine scale, NSST reconstruction was used on three different scales, to obtain twelve transform-based features for four spectral bands.

(e) Sobel operator was used to obtain four edge-based features for four spectral bands.

(f) Ten vegetation indexes (NDVI, NDWI, GR, DVI, RVI, SAVI, OSAVI, MSAVI, PVI, and EVI) were also extracted from four spectral bands, the formula of these vegetation indexes is listed in Table 2.

**Table 2.** The formula of vegetation indexes.

| Vegetation Indexes | Formula | Parameter Explanation |
|---|---|---|
| NDVI | $NDVI = (NIR - R)/(NIR + R)$ | NIR is the near-infrared band and R is the red band |
| NDWI | $NDWI = (G - NIR)/(G + NIR)$ | G is the green band |
| GR | $GR = G/(B + G + R)$ | B is the blue band |
| DVI | $DVI = NIR - R$ | |
| RVI | $RVI = NIR/R$ | |
| SAVI | $SAVI = \frac{NIR - R}{NIR + R + p}(1 + p)$ | p is the percent of vegetation cover |
| OSAVI | $OSAVI = \frac{(1 + 0.16)(NIR - R)}{(NIR + R + 0.16)}$ | |
| MSAVI | $MSAVI = \sqrt{(NIR + 0.5)^2 - 2(NIR - R)}$ | |
| PVI | $PVI = \frac{NIR - aR - b}{\sqrt{1 + a^2}}$ | $a = 10.489,\ b = 6.604$ |
| EVI | $EVI = g \times \frac{NIR - R}{NIR + C_1 R - C_2 B + L}$ | $g = 2.5,\ C_1 = 6,\ C_2 = 7.5,\ L = 1$ |

Together 303 features were extracted from ZY-3 images, as shown in Table 3.

**Table 3.** Image features extracted from ZY-3 images.

| Feature Types | Image Features | Number |
|---|---|---|
| Spectral-based | blue, green, red, near-infrared bands, brightness | 5 |
| GLCM-based | mean, variance, homogeneity, dissimilarity, contrast, entropy, angular second moment, correlation | 192 |
| Morphological-based | morphological profiles (opening and closing), different morphological profiles (opening and closing) | 80 |
| Transform-based | reconstructed features using non-subsampling shearlet transform | 12 |
| Edge-based | Sobel edge feature | 4 |
| Vegetation indexes | NDVI, NDWI, GR, DVI, RVI, SAVI, OSAVI, MSAVI, PVI, EVI | 10 |
| Total | | 303 |

### 3.2. Feature Hierarchical Selection

Variable importance (VI) of image features was calculated by random forest algorithm, as shown in Equation (1):

$$VI = \sum_{t=1}^{N_{tree}} \frac{E_{OOB2} - E_{OOB1}}{N_{tree}} \tag{1}$$

where $E_{OOB1}$ is the out-of-bag (OOB) error of decision trees, $E_{OOB2}$ is the OOB error when the image feature $F_i$ is replaced, $N_{tree}$ is the number of decision trees. After the image

feature $F_i$ is replaced, if the *OOB* error changes obviously, it indicates that the image feature $F_i$ is more important.

Assuming that the data sequence is $\{X_1, X_2, X_3, \cdots, X_n\}$, the statistic that obeys the overall distribution is $F(x, \theta)$, the number is $n$, the expectation of the data sequence $\{X_1, X_2, X_3, \cdots, X_n\}$ is $\widetilde{u}$. $\widetilde{u}_k$ is the point estimate that only depends on the expectation $\widetilde{u}$, shown as Equation (2), in which $x_i$ is the *i*th statistic value, and $x_k$ is the kth statistic value.

$$\widetilde{u}_k = \frac{\sum\limits_{i=1}^{k} x_i + (n-k)x_k}{k} \tag{2}$$

In this paper, the jumping degree $t_k$ was designed for feature hierarchical selection. Firstly, all the extracted image features were sorted in ascending order according to their VI values. Secondly, the jumping degree of each image feature was calculated, shown as Equations (3) and (4).

$$t_k = \frac{\widetilde{u}_{k+1}}{\widetilde{u}_k} \tag{3}$$

$$\widetilde{u}_k = \frac{\sum\limits_{i=1}^{k} s_i + (n-k)s_k}{k} \tag{4}$$

where $t_k$ is the jumping degree at point k, $\widetilde{u}_k$ is point estimate at point k, $\widetilde{u}_{k+1}$ is point estimate at point (k+1), $s_i$ is VI value of image features $F_i$, $s_k$ is VI value of image features $F_k$, and $n$ is the number of image features.

Finally, if $t_k(k \geq 2)$ is greater than all the jumping degrees of its previous $(k-1)$ image features, $k$ is regarded as the point that distinguishes different feature groups, and the previous $(k-1)$ image features belong to the same feature group. The remaining image features after the removal of the previous $(k-1)$ image features repeat the above rule, until all image features are distinguished hierarchically.

*3.3. Saliency-Guided Binary Change Detection*

Dynamic wetlands monitoring in large-scale areas was an effective method that firstly extracted changed and unchanged areas, and then identified different change categories. Changed and unchanged areas were extracted using binary CD methods, and different change categories were identified using multiple CD methods. Many binary CD methods have been proposed, they can be divided into pixel-based, sub-pixel-based, object-based, and scene-based CD methods, according to different analysis units [35].

Pixel-based and object-based binary CD methods are widely used, but each of them has its advantages and disadvantages. Pixel-based CD methods are sensitive to image registration errors, and their salt-and-pepper phenomena are serious. Object-based CD methods can improve salt and pepper phenomena, and they are less affected by image registration errors, but they are greatly affected by image segmentation parameters. In order to utilize their advantages and solve their disadvantages, we previously proposed a saliency-guided binary CD method combining pixel-based and object-based CD methods, and describe it as follows. Firstly, the different images are obtained and saliency detection is used to generate saliency maps of the different images using a maximum symmetric surround (MSS) algorithm. Then, the combination of fuzzy C-means (FCM) with Markov random field (MRF) is used to extract the initial CD result at a pixel-based level. Secondly, a multi-scale segmentation algorithm is utilized for object-oriented image segmentation, in which the rate of change of local variance (ROC-LV) is used to estimate the optimal segmentation scales. Finally, the uncertainty index of segmentation objects is constructed to adaptively select changed and unchanged samples, and these samples are then used for training random forest classifier, to obtain the final CD results. The reference paper [36] can be consulted for further details.

### 3.4. Change Identification Using Sample Transfer Learning

For large-scale areas with complex land use and land cover classes, training sample collection is time-consuming and labor-intensive, and only small amounts of training samples are available at only one of the pre- and post-changed times. Therefore, training sample transfer learning was designed for change identification in order to obtain the "from-to" changes. Its flowchart is described in Figure 2.

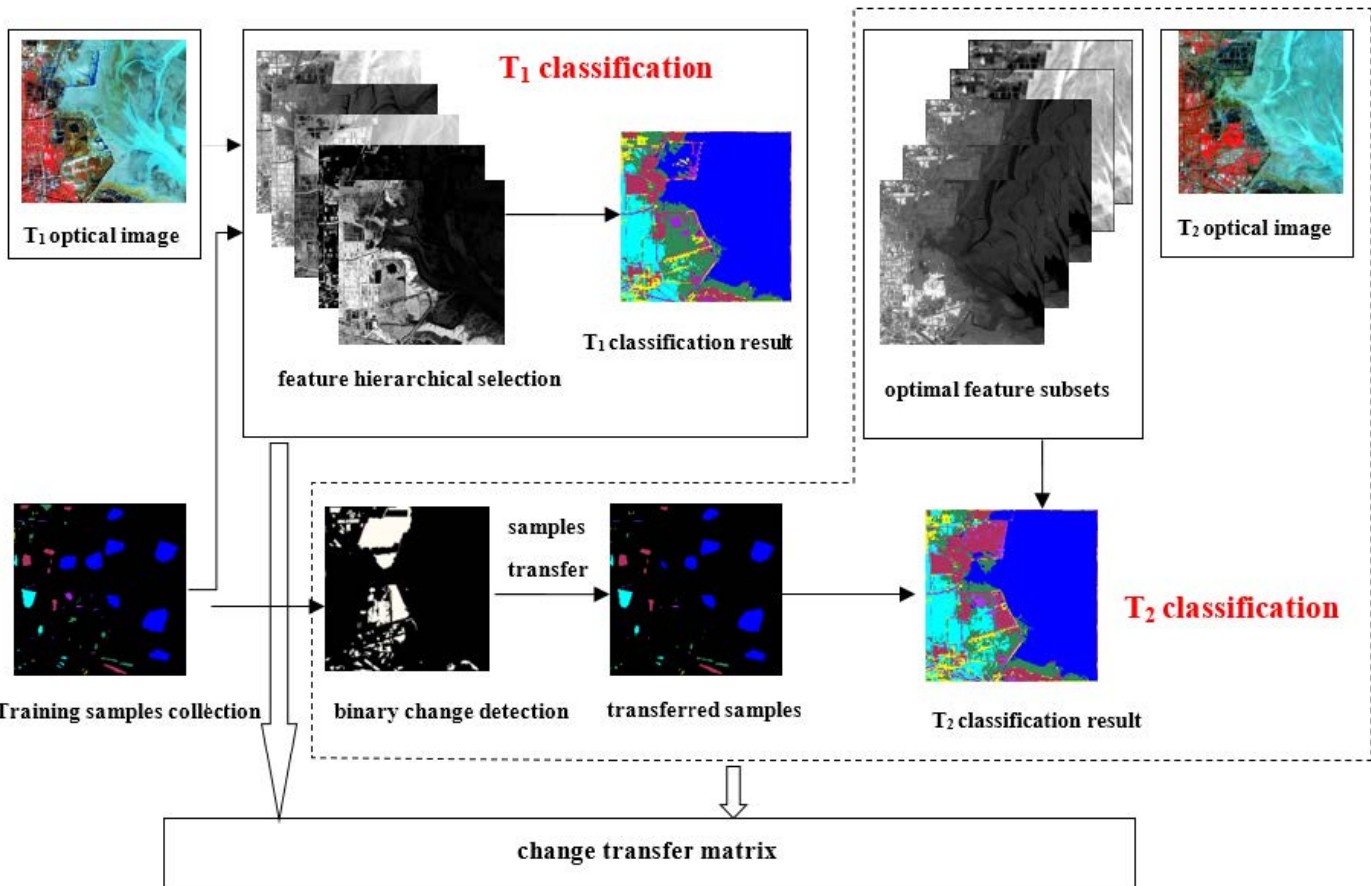

**Figure 2.** Change identification using feature hierarchical selection and training samples transfer learning methods.

For optical images at $t_1$ time, image features were optimized using the feature hierarchical selection method, and optimal feature subsets were then input into random forest classifier in order to obtain different wetland distributions at $t_1$. Six different training samples were collected using unmanned aerial vehicle (UAV) flights. From $t_1$ to $t_2$, some training samples changed from one wetland class to another one. These changed training samples could not be training samples at $t_2$ time, instead only unchanged training samples were transferred as training samples at $t_2$. Transferred training samples were used for random forest classification at $t_2$ time, and only image features belonging to optimal feature subsets were extracted at $t_2$ time. Different wetland distributions at $t_2$ were obtained by random forest classification, in which wetland classes in unchanged areas were the same. Wetland distribution in changed areas at $t_1$ and $t_2$ time was compared with each other and the change transfer matrices were obtained.

## 4. Results

### 4.1. Feature Selection Results

VI values of 303 image features of ZY-3 data in 2013 were calculated using random forest and were sorted in ascending order. According to the feature hierarchical selection

method, features with similar VI values were divided into the same feature group. The red dots in Figure 3 are the points distinguishing different feature groups, corresponding to the 35, 71, 109, 158, 225, 265, and 281st features. In this way, the 303 features were divided into eight feature groups, represented as $F = \{f_1, f_2, f_3, f_4, f_5, f_6, f_7, f_8\}$, in which $f_1 = [1 \sim 35]$, $f_2 = [36 \sim 71]$, $f_3 = [72 \sim 109]$, $f_4 = [110 \sim 158]$, $f_5 = [159 \sim 225]$, $f_6 = [226 \sim 265]$, $f_7 = [266 \sim 281]$, and $f_8 = [282 \sim 303]$.

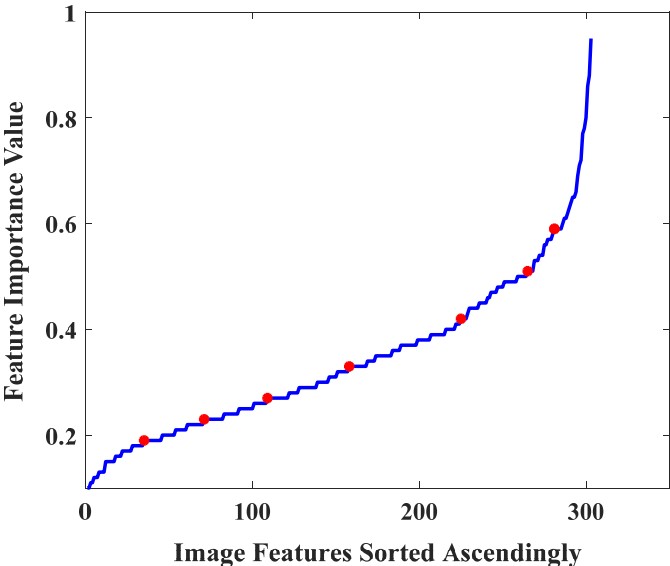

**Figure 3.** Hierarchical features in which the red dots are the points distinguishing different feature groups.

Ground reference sample data of six different wetland categories were collected using UAV flight, including sea, open water, farmland, aquaculture pond, salt marsh, and building. In this case, small amounts of training samples in 2013 were obtained, and they were used for wetland classification in 2013. In order to verify the influence of different feature combinations on wetland classification, different feature subsets were input into random forest classifiers for wetland classification. Overall accuracy and Kappa coefficient of different feature subsets are listed in Table 4. Generally, the overall accuracy of wetland classification decreased sequentially, along with the feature groups with low VI values being deleted sequentially. The feature groups that are beneficial to wetland identification were deleted, which reduced the distinction of wetland classes. The overall accuracy of wetland classification increased when the feature groups $\{f_1\}$ were deleted. The overall accuracy of $F_1 = \{f_2, f_3, f_4, f_5, f_6, f_7, f_8\}$ for wetland identification was the highest, at 98.51%. The Kappa coefficient of $F_1 = \{f_2, f_3, f_4, f_5, f_6, f_7, f_8\}$ was 0.9763, and was also the highest.

**Table 4.** The difference of wetland classification using different feature subsets.

| Different Feature Subsets | Overall Accuracy (%) | Kappa Coefficient |
|---|---|---|
| $F = \{f_1, f_2, f_3, f_4, f_5, f_6, f_7, f_8\}$ | 98.36 | 0.9731 |
| $F_1 = \{f_2, f_3, f_4, f_5, f_6, f_7, f_8\}$ | 98.51 | 0.9763 |
| $F_2 = \{f_3, f_4, f_5, f_6, f_7, f_8\}$ | 98.17 | 0.9700 |
| $F_3 = \{f_4, f_5, f_6, f_7, f_8\}$ | 98.24 | 0.9711 |
| $F_4 = \{f_5, f_6, f_7, f_8\}$ | 97.99 | 0.9670 |
| $F_5 = \{f_6, f_7, f_8\}$ | 97.59 | 0.9605 |
| $F_6 = \{f_7, f_8\}$ | 96.57 | 0.9439 |
| $F_7 = \{f_8\}$ | 96.07 | 0.9359 |

### 4.2. Change Identification Results

Based on ZY-3 images in 2013 and 2018, our previously proposed binary CD method could obtain the unchanged and changed areas in the study area. The overall accuracy of the proposed CD method was 93.51% [36] and it could meet the reliability of subsequent wetland change identification. According to the flowchart of change identification shown in Figure 2, wetland classification results in 2018 could be obtained. The wetland classification results in 2013 and 2018 are shown in Figure 4, the overall accuracy was 98.51% and 91.16%, respectively. This could meet the requirement of actual wetland monitoring.

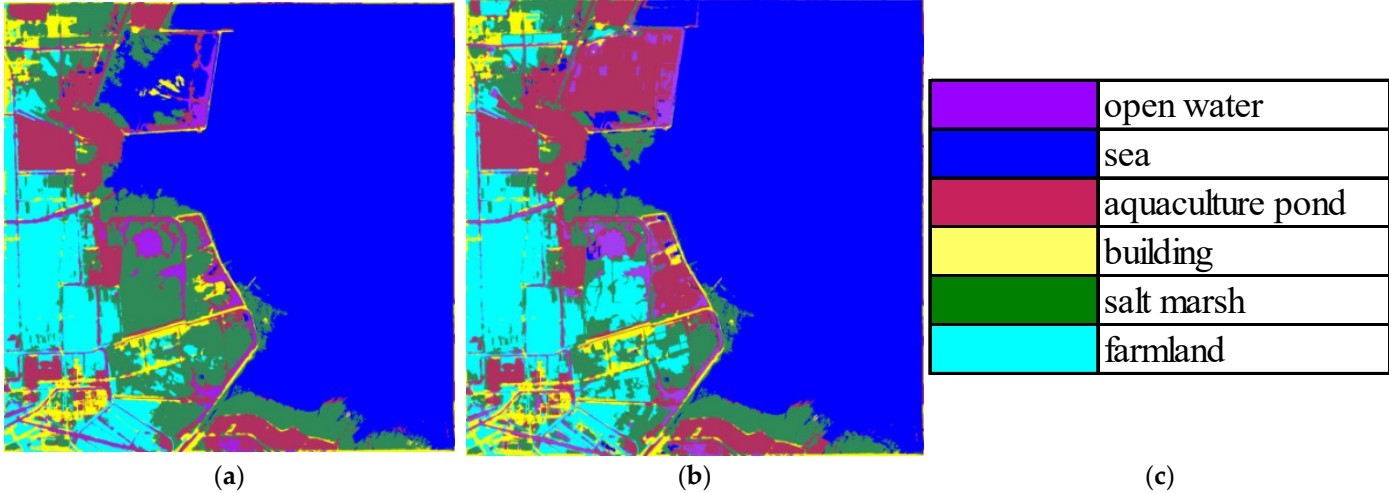

**Figure 4.** Wetland classification results: (**a**) 2013; (**b**) 2018; (**c**) wetland classes.

The change transfer matrix between six different wetland categories from 2013 to 2018 is shown in Table 5. The increase of salt marsh was mainly from the sea area (2.27 km$^2$) because salt marsh had strong adaptability and fertility and could expand rapidly throughout coastal areas. The increase of the aquaculture pond was from the sea area (10.58 km$^2$) and salt marsh (5.24 km$^2$) because the aquacultural industry had considerable economic benefits. Some new aquaculture ponds emerged under a reclamation project, and some salt marshes in original natural landscapes were developed as aquaculture ponds. The increase in farmland was from salt marshes because the agricultural industry also developed in coastal areas. The increase of building was from salt marshes and farmland because of population migration to wetland areas which occupied agricultural land and salt marshes to build residential and industrial land.

**Table 5.** Change transfer matrix between different wetland categories from 2013 to 2018 (units: km$^2$).

| 2013＼2018 | Sea | Aquaculture Pond | Salt Marsh | Farmland | Open Water | Building | Total |
|---|---|---|---|---|---|---|---|
| Sea | 199.52 | 10.58 | 2.27 | 0.02 | 1.34 | 0.06 | 213.79 |
| Aquaculture pond | 0.23 | 40.5 | 0.19 | 1.08 | 0.59 | 0.22 | 42.82 |
| Salt marsh | 0.35 | 5.24 | 50.61 | 10.27 | 1.51 | 0.5 | 68.49 |
| Farmland | 0.09 | 0.38 | 0.62 | 40.66 | 0.52 | 0.88 | 43.14 |
| Open water | 0.07 | 0.93 | 0.03 | 0.04 | 10.39 | 0.03 | 11.49 |
| Building | 0.2 | 1.34 | 0.2 | 0.36 | 0.2 | 17.96 | 20.26 |
| Total | 200.46 | 58.98 | 53.91 | 52.44 | 14.56 | 19.65 | 400 |

### 5. Conclusions

High-resolution optical data can observe the changes in coastal wetlands in detail. Image features can be extracted from these high-resolution data, but the impact of different features on change identification is not clear, and the combination of a large number of features could easily cause a "dimension disaster" problem. Therefore, the feature

hierarchical selection method was proposed taking into account the jumping degree of different image features. The influence of different features on wetland classification was analyzed and those features which had a slight influence on wetland classification were deleted to not only reduce dimensionally but also to obtain optimal features subsets.

In addition, only small amounts of training samples were accessible at pre- or post-changed time. Therefore, the training samples transfer learning strategy was designed for wetland classification, and the classification results at pre- and post-changed times were compared to obtain a change transfer matrix. The main conclusions are as follows:

(1) The jumping degree was introduced to design a feature hierarchical strategy in order to obtain optimal feature subsets. The feature selection results showed that the feature hierarchical selection method could provide a quantitative reference for optimal feature subsets selection.

(2) The training samples transfer learning strategy was used to classify post-changed optical data without recollecting training samples. It could obviously save the effort of collecting training samples. The overall accuracy of the transferred training samples was 91.16%, demonstrating that it could ensure the accuracy requirements for change monitoring.

(3) The southeastern coastal wetlands located in Jiangsu Province were used as a study area and ZY-3 images in 2013 and 2018 were used to conduct experiments. The results demonstrated that salt marshes increased mainly from the sea area (2.27 km$^2$) because salt marshes expand rapidly throughout coastal areas and aquaculture ponds increased from the sea area (10.58 km$^2$) and salt marshes (5.24 km$^2$) because of the considerable economic benefits of the aquacultural industry.

**Author Contributions:** Conceptualization, R.W. and J.W.; methodology, R.W.; software, R.W.; validation, R.W. and J.W.; formal analysis, R.W.; investigation, R.W. and J.W.; data curation, R.W.; writing—original draft preparation, R.W.; writing—review and editing, R.W. and J.W.; visualization, R.W.; supervision, J.W. All authors have read and agreed to the published version of the manuscript.

**Funding:** The work was supported by the National Natural Science Foundation of China under Grant 41871203, a grant from the State Key Laboratory of Resources and Environmental Information System under Grant 2022-30, Key Laboratory for Digital Land and Resources of Jiangxi Province, East China University of Technology under Grant DLLJ202008, Key Laboratory of Surveying and Mapping Science and Geospatial Information Technology of Ministry of Natural Resources under Grant 2020-3-6, and the Scientific Research Innovation Team Project of Neijiang Normal University under Grant 2021TD01.

**Institutional Review Board Statement:** Not applicable.

**Informed Consent Statement:** Not applicable.

**Data Availability Statement:** Data are contained in this article.

**Acknowledgments:** The authors would like to thank the reviewers for their help on the article.

**Conflicts of Interest:** The authors declare no conflict of interest.

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
