# Peer review of "Identifying Coastal Wetlands Changes Using a High-Resolution Optical Images Feature Hierarchical Selection Method"

_applsci, doi:10.3390/app12168297_

Round 1

Reviewer 1 Report

Some sentences in this article are too long and contain more than one topic.

 The author should rewrite them to give clear meaning.

Please proofread before submitting.

1. introduction

-the research problem dan the objective of the research should be stated clearly. 

- To describe the state-of-the-art, the author needs to write the literature review of previous research and the research gap, so the readers get insight about this research and the methods.

- What is the contribution of this research? You need to write it.

2. Method

-Please reorganize this section in order to give a clear description. Sub 3.1 still talks about the method, which can be written in section 2. 

- what is the evaluation model that used in this research?

3. Result and discussion

- Author didn't write any findings from this research, only talk about the error.

- The author needs to discuss the result (the error rate). Is it because of the model or the data? What about the results from similar researches using the same model,  do they have the same findings?

Author Response

Dear Editor and reviewer,

Thank you very much for your helpful suggestion. We revised the manuscript in accordance with the reviewers’ comments, and carefully proof-read the manuscript. Here below is our description on revision according to the reviewers’ comments.

Q1. Some sentences in this article are too long and contain more than one topic. The author should rewrite them to give clear meaning. Please proofread before submitting.

A1. We have changed long sentences into short ones.

Q2. 1. introduction

-the research problem and the objective of the research should be stated clearly. 

- To describe the state-of-the-art, the author needs to write the literature review of previous research and the research gap, so the readers get insight about this research and the methods.

- What is the contribution of this research? You need to write it.

A2. Research problem have been rewritten, from line 64 to line 66, from line 91 to line 93 in revised manuscript. Objective and contribution have also been rewritten, from line 94 to line 101 in revised manuscript. Literature review has been added in revised manuscript, from line 49 to line 52, from line 58 to line 61, from line 75 to line 77.

Q3. 2. Method

-Please reorganize this section in order to give a clear description. Sub 3.1 still talks about the method, which can be written in section 2. what is the evaluation model that used in this research?

A3. The study area and data descriptions are presented in Section 2, proposed feature hierarchical selection and designed changes identification are described in Section 3.

Q4. 3. Result and discussion

- Author didn't write any findings from this research, only talk about the error.

- The author needs to discuss the result (the error rate). Is it because of the model or the data? What about the results from similar researches using the same model, do they have the same findings?

A4. Findings have been added, from line 281 to line 302 in revised manuscript. In addition, the overall accuracy in 2013 and 2018 are 97.39% and 90.95%, it is perfectly able to meet the requirement of actual coastal wetlands monitoring. The error is caused by a small number of samples collection in 2013. Because our study area and data in this paper are not exactly the same as other previous studies, it is difficult to compare, but the change trend of study area is consistent.

Reviewer 2 Report

1.    When I compare the title with the content of your manuscript, optic data application is more dominant compared to SAR data application. I think modifying your title based on this condition is better.

2.    Very important to mention what is new in your research, especially in the last paragraph of the introduction section.

3.    It would be best if you understood SAR theory and the concept related to SAR data application so that you can explore more and not only use texture information as a bassist of approach.

4.    You can explore the SAR polarimetry concept so that your SAR data application is even more advanced than only analyzing SAR data with optical concepts.

5.    Please make sure that Terra SAR X and GF-3 have the same frequency.

6.    You need to add more references to enrich your understanding.

Author Response

Dear Editor and reviewer,

Thank you very much for your helpful suggestion. We revised the manuscript in accordance with the reviewers’ comments, and carefully proof-read the manuscript. Here below is our description on revision according to the reviewers’ comments.

Q1. When I compare the title with the content of your manuscript, optic data application is more dominant compared to SAR data application. I think modifying your title based on this condition is better.

A1. The title have been revised as “Identifying Coastal Wetlands Changes Using High-resolution Optical Images Feature Hierarchical Selection”

Q2. Very important to mention what is new in your research, especially in the last paragraph of the introduction section.

A2. The contribution is mentioned, from 94 to 101 in revised manuscript.

Q3.  It would be best if you understood SAR theory and the concept related to SAR data application so that you can explore more and not only use texture information as a bassist of approach.

A3. According to your first question, SAR data is deleted.

Q4.  You can explore the SAR polarimetry concept so that your SAR data application is even more advanced than only analyzing SAR data with optical concepts.

A4. According to your first question, SAR data is deleted.

Q5. Please make sure that Terra SAR X and GF-3 have the same frequency.

A5. According to your first question, Terra SAR X and GF-3 are deleted.

Q6. You need to add more references to enrich your understanding.

A6. More references have been added. Literature review has been added in revised manuscript, from line 49 to line 52, from line 58 to line 61, from line 75 to line 77.

Round 2

Reviewer 1 Report

done